# An Insight into the Roles of MicroRNAs and Exosomes in Sarcoma

**DOI:** 10.3390/cancers11030428

**Published:** 2019-03-26

**Authors:** Isaku Kohama, Nobuyoshi Kosaka, Hirotaka Chikuda, Takahiro Ochiya

**Affiliations:** 1Division of Molecular and Cellular Medicine, National Cancer Center Research Institute, 5-1-1 Tsukiji, Chuo-ku, Tokyo 104-0045, Japan; ikohama@ncc.go.jp; 2Department of Orthopaedic Surgery, Gunma University Graduate School of Medicine, 3-39-22 Showamachi, Maebashi, Gunma 371-8511, Japan; chikudah@gunma-u.ac.jp; 3Department of Molecular and Cellular Medicine, Institute of Medical Science, Tokyo Medical University, 6-1-1 Shinjuku, Shinjuku-ku, Tokyo 160-8402, Japan; 4Department of Translational Research for Extracellular Vesicles, Tokyo Medical University, 6-1-1 Shinjuku, Shinjuku-ku, Tokyo 160-8402, Japan

**Keywords:** cancer, sarcoma, microRNA, exosome, metastasis

## Abstract

Sarcomas are rare solid tumors, but at least one-third of patients with sarcoma die from tumor-related disease. MicroRNA (miRNA) is a noncoding RNA that regulates gene expression in all cells and plays a key role in the progression of cancers. Recently, it was identified that miRNAs are transferred between cells by enclosure in extracellular vesicles, especially exosomes. The exosome is a 100 nm-sized membraned vesicle that is secreted by many kinds of cells and contains miRNA, mRNA, DNA, and proteins. Cancer uses exosomes to influence not only the tumor microenvironment but also the distant organ to create a premetastatic niche. The progression of sarcoma is also regulated by miRNAs and exosomes. These miRNAs and exosomes can be targeted as biomarkers and treatments. In this review, we summarize the studies of miRNA and exosomes in sarcoma.

## 1. Introduction

From the 1970s to the 1980s, the results of treatment for sarcoma has greatly improved by the new entrants doxorubicin, ifosfamide, high-dose methotrexate, and cisplatin [1]. However, sarcoma treatment has not been improved for almost 30 years. The rarity and diversity of sarcomas are part of the reason preventing the development of treatments for sarcoma. The incidence of sarcoma is far less (i.e., one-hundredth) than that of carcinoma. Moreover, there are more than 50 subtypes of sarcoma, and the genetic backgrounds and treatments are different. Similar to other cancers, metastasis is the major cause of death for patients with sarcoma. One-third of the patients with sarcoma die because of metastasis [2].

MicroRNA (miRNA) is a type of noncoding RNA (ncRNA) that consists of 17–25 base pairs of short ncRNA and regulates gene expression by suppressing translation or enhancing mRNA degradation via targeting the 3′-UTR, which is post transcriptional regulation of gene expression [3]. It is well known that miRNAs regulate biological homeostasis and pathological processes, including cancers. miRNA also plays a key role in cancer progression, including sarcoma. Furthermore, several miRNAs are encapsulated by exosomes [4]. It has been found that cancer-derived exosomes containing miRNAs are transferred to surrounding cells in microenvironments and distant organs. These miRNAs have attracted attention as targets for biomarkers and treatment.

In this review, we discuss the production, general synthesis, and function of miRNAs and exosomes. Furthermore, we summarize the miRNA and exosome regulation of sarcomas.

## 2. MicroRNA

With the technological innovation of the next-generation sequencer, it has become clear that a large number of ncRNAs are transcribed. RNA is divided into a messenger RNA (mRNA) containing information to make protein and a ncRNA that does not encode a protein. miRNA is a type of ncRNA and is synthesized and processed in the nucleus and cytoplasm (Figure 1A) [5,6,7].

miRNA plays a key role in various physiological functions. The expression of miR-146 and miR-155 is upregulated in response to inflammatory stimuli such as Toll-like receptor (TLR) ligands in myeloid cells [8,9] These miRNAs are part of the negative feedback loop of inflammatory pathways in myeloid cells. miR-146 and miR-155 suppress the activation of the TRL pathway by directly targeting IL-1R-associated kinase 1 (IRAK1), TNFR-associated factor 6 (TRAF6), and TAK1-associated binding protein 2 (TAB2), which are downstream of TLR. miR-126 regulates vascular endothelial cells and vascular development. miR-126 directly suppresses the expression of vascular cell adhesion molecule 1 (VCAM-1) in endothelial cells [10]. VCAM-1 is an adhesion molecule that is induced by inflammatory cytokines. When miR-126 expression is suppressed in HUVECs, the induction of VCAM-1 expression by TNFα is promoted. In addition, miR-126 plays an important role in the maintenance of vascular integrity during embryogenesis [11]. Approximately 12% of miR-126^−/−^ mice died by postnatal day 1 because of leaky vessels and hemorrhaging.

Sarcoma is a neoplasm derived from mesenchymal tissue such as bone and adipose tissue. Bone is continually being remodeled by osteoclasts and osteoblasts, which are selectively differentiated from hematopoietic and mesenchymal stem linage cells [12]. Several miRNAs are involved in the induction of osteoclast, osteoblast, and adipocyte differentiation (Figure 2). In osteoclasts, hematopoietic stem cells differentiate into monocytes and become osteoblastic precursors under the influence of PU.1, microphthalmia-associated transcription factor (MITF), and macrophage colony-stimulating factor (M-CSF). Osteoblastic precursors mature into osteoclasts by the interaction of receptor activator of nuclear factor kappa-B (RANK) and RANK ligand (RANKL). miR-233 and miR-29 prevent osteoclastic maturity by targeting nuclear factor I/A (NFIA), which is the activator of M-CSF [13,14]. miR-155 promotes the differentiation of monocytes into macrophage and osteoclast lineage cells [15]. miR-155 suppresses MITF expression and inhibits the differentiation of monocytes into osteoclasts. On the other hand, osteoblasts are derived from mesenchymal stem cells (MSCs) and differentiate into osteocytes through pre-osteoblasts and mature osteoblasts. Runt-related transcription factor 2 (Runx2) is expressed from the early stage of differentiation, and ATF 4 is expressed in the later stage. Runx2 is a key transcription factor associated with osteoblast and chondrocyte differentiation [16]. miR-133 directly targets Runx2 and inhibits osteogenic differentiation, while, high miR-214 expression is correlated with a reduction in bone formation, which targets ATF4 and inhibits osteoblast activity [17]. Adipocytes are differentiated from MSCs. CCAAT-enhancer-binding proteins (C/EBPs) and proliferator-activated receptor γ (PPARγ) play a key role in adipocyte differentiation [18]. miR-27a/b and miR-33b directly target PPARγ and suppress adipocyte differentiation [19,20,21]. miR-33b, miR-25, and miR-342-3p target C/EBPα, which is a key promoter of adipocyte differentiation [21,22,23].

It was reported in 2007 that miRNA is released in extracellular vesicles (EVs) [24]. In addition, it was successively reported that exosomal miRNAs are transferred to other cells and regulate the progression of various diseases, including cancer [25,26,27]. After these findings, accumulating evidence indicates that miRNA regulates gene expression in other cells via exosomes. In the next section, we will introduce the biogenesis and functions of exosomes in cancer.

## 3. Exosome and Exosomal miRNA

The exosome is a small vesicle approximately 100 nm in size that contains DNA, RNA, and protein. Exosomes are members of the EV family, and EVs are roughly classified into three types depending on the production route and size: Exosomes (100 nm), microvesicles (MVs; 100–1000 nm), and apoptotic bodies (800–5000 nm) (Figure 1B) [28,29]. The exosome plays a key role in intercellular communication and is released from almost all types of cells, including normal cells and cancer cells. miRNAs are contained in exosomes and are transferred to the other cells and distant organs (Figure 1C) [5]. For instance, bone marrow-derived MSCs (BM-MSCs) induce stemness phenotypes and drug resistance in colorectal cancer cells (CRC) [30]. In the exosomes from BM-MSCs, miR-142-3p is highly expressed and transferred to CRC and targets Numb, which is an inhibitor of Notch signaling. The expression of Numb is downregulated but the markers of the Notch signaling pathway are upregulated by exosomal miR-142-3p. On the other hand, the exosome of breast cancer (BC) promotes brain metastasis. The exosomal miR-181c from BC disturbs the tight junction of the blood–brain barrier, which promotes brain metastasis [31]. Bone is the site to which BC tends to metastasize. BC relapse has been known to occur 10 years after resection of the primary tumor [32]. The exosomes derived from BM-MSCs were found to suppress the proliferation of a bone metastatic BC cell line but decreased its sensitivity to docetaxel [33]. In addition, exosomal miR-222/223 derived from MSCs promoted quiescence in BC cells [34].

It has been shown that exosomal miRNA is a promising candidate biomarker. Exosomes can be detected in body fluids, such as blood and urine [35,36]. The seven serum exosomal miRNAs (let-7a, miR-1229, miR-1246, miR-150, miR-21, miR-223, and miR-23a) were found to be significantly higher in patients with colorectal cancer than in healthy controls [37]. The expression levels of miR-21 and miR-1246 were significantly higher in plasma-derived exosomes from BC patients than in those from healthy controls [38]. In addition, the expression levels of miR-21, miR-375, and let-7c in urinary exosomes were significantly higher in patients with prostate cancer than in healthy controls [39].

Sarcoma has different genetic backgrounds for each subtype. Several sarcomas harbor specific fusion genes, such as EWS-FLI1 of Ewing’s sarcoma and PAX3/7-FOXO1 of rhabdomyosarcoma. In the next section, we show the regulation of miRNAs and exosomes in sarcoma according to each genetic feature.

## 4. The Functions of miRNA and Exosomes in Sarcoma

### 4.1. Osteosarcoma

Osteosarcoma (OS) is the most common primary sarcoma of bone. The predilection age is 10–14 years old, which is consistent with the age of bone growth [40]. The mainstay of treatment for OS is surgery. However, the survival of patients treated with only surgery is less than 20% [41,42]. In the early 1970s, the induction of multidrug chemotherapy with surgery greatly improved the treatment results. Current therapies cure more than 70% of OS patients [43]. However, survival for patients with metastasis and relapse has not improved for almost 30 years.

Conventional high-grade OS is a genomically unstable tumor with complex aneuploid karyotypes and structural chromosomal aberrations [44,45,46]. Recurrent amplification or loss at several distinct chromosomal regions has been detected [44,47,48]. Each region includes gene regions related to tumor development and osteogenesis (Table 1). In particular, the amplification of the 6p12–21 and 17p11 regions is detected most frequently, representing approximately 30–60% of the OS samples. Runx2 is a gene located in 6p12–p21. Runx2 is highly expressed in OS tumors, and Runx2 expression is associated with poor prognosis. When Runx2 expression is downregulated in OS cells, the sensitivity of doxorubicin is improved by activating apoptosis [49]. Several miRNAs target Runx2, which regulates the differentiation of osteoblasts and chondroblasts [12]. Among them, miR-34c expression is regulated by p53 and suppresses Runx2 expression, which in turn suppresses the cell proliferation of OS [50]. miR-338-3p, miR-23a, and miR-203 are downregulated in OS cells and tissues. These miRNAs target Runx2 directly and suppress cell proliferation and the ability of invasion and migration [51,52,53]. Next, it has been reported that the expression of vascular endothelial growth factor (VEGF) and VEGF receptor (VEGFR) are related to prognosis and that circulating VEGF levels are associated with the development of lung metastasis in OS [54,55]. VEGF expression also indicates a positive correlation with drug resistance and tumor microvessel density (MVD) [56,57]. There are conflicting reports about the relationship between MVD density and OS prognosis, but it is necessary to verify this relationship by continuing research [58,59,60]. miR-134, miR-145, and miR-20b are downregulated in OS tissues and cells [61,62,63]. They suppress VEGF and VEGFR expression. miR-20b targets hypoxia inducible factor 1 α (HIF1α) and suppresses VEGF expression by suppressing HIF1α expression. On the other hand, miR-337-5p is highly expressed in OS tissue. The high expression of miR-337-5p is significantly related to poor prognosis and lung metastasis. The overexpression of miR-337-5p in OS cells promotes cell growth by activating VEGF, erythroblastic leukemia viral oncogene homolog (ERBB), and mitogen-activated protein kinase (MAPK) pathways [64]. Depletion of cell division cycle 5-like (CDC5L) causes mitotic arrest, chromosome misalignments, and leads to mitotic catastrophe [65]. Mohammadi et al. reported that CDC5L mRNA levels in OS tissues are higher than those in adjacent normal tissues [66]. High CDC5L expression is correlated with an advanced TNM stage. Since miRNAs that regulate CDC5L have not been reported, this focus can be expected for future research.

MAPK 7 and MAP2K4 are genes located in 17p11. MAPK is induced by many kinds of extracellular stimuli, such as mitogen-activated factors, osmotic stress, heat shock, and proinflammatory cytokines [81]. These high expression levels are related to the poor prognosis of OS [82]. MAPK7 promotes cell growth and invasion and migration in OS cells [83,84]. miR-143 and miR-125b directly targets MAPK7 and suppresses MAPK7 expression in OS cells. These miRNAs suppress cell growth and invasion and migration [67,68]. Cyclin-dependent kinase 4 (CDK4) and mouse double minute 2 homolog (MDM2) are genes located in 12q13–q14. CDK4 and MDM2 gene amplifications have been found in 7–8% of OS patients [85]. In addition, high CDK4 expression is related to poor prognosis and metastasis of OS [86]. miR-506-3p suppresses the expression of CDK4 and matrix metalloproteinase (MMP) 9 by targeting ras-related protein rab-3D (RABD3D) [69]. Myc is an oncogene at 8q24. Myc is activated by mitogenic signals such as Wnt, Notch, and receptor tyrosine kinase (RTK) signaling. The expression of c-Myc in relapse and metastatic tumors of OS is higher than that in the primary tumor. In addition, a high expression of c-Myc shows a positive relationship with later metastasis [87]. miR-33b, miR-449c, and miR-135b directly target c-Myc. Their expression is suppressed in OS tissue and cell lines. When these miRNAs are overexpressed in OS cells, cell growth, migration, and invasion are suppressed [70,71,72]. miR-214, miR-107, and miR-137 regulate c-Myc via Wnt signaling. miR-107 and miR-137 target Dickkopf-related protein 1 (DKK1) and Fibulin 4, respectively [73,74,75].

Somatic mutations in TP53 and Retinoblastoma 1 (RB1) are most frequently reported [88,89]. Li–Fraumeni syndrome (LFS) is a cancer predisposition syndrome caused by a germline mutation of the TP53 gene. Individuals with LFS are known to have a high risk of developing osteosarcoma [90]. miR-381 and miR-373 affect p53 regulation via Wnt and PI3K/AKT signaling [76,77]. miR-1281 is upregulated by endoplasmic reticulum (ER) stress. miR-1281 binds to the promoter region of TP53 [78]. p63 is a member of the p53 family and has been linked to tumorigenesis [91]. ⊿Np63α, which lacks the N-terminal transactivation domain of p63, suppresses the expression of miR-527 and miR-665 in OS cells, which promotes lung metastasis in vivo via activating the transforming growth factor-β (TGF-β) pathway [79]. Patents with hereditary retinoblastoma are at a high risk of developing osteosarcoma [92]. miR-142 phosphorylates Rb and promotes apoptosis in OS cells [80].

Notch signaling has a pivotal role not only in embryo development but also in the regulation of cancer [93]. Notch signaling contributes to the metastatic capability and malignancy of OS [94]. miR-26a directly targets Jaggd1, which is a ligand of Notch, and maintains the stemness of OS. The expression of miR-26a was found to be significantly suppressed in a population of cancer stem cells (CSCs) of OS. When miR-26a is overexpressed in OS cells, the activation of Notch signaling is suppressed and inhibits aggressiveness in vitro and in vivo. In addition, the low expression of miR-26a in OS tissue is related to a poor prognosis [95]. Diallyl trisulfide (DATS) is a natural compound derived from allium vegetables that reduces the risk of cardiovascular disease and shows anticancer effects. When OS cells are treated with DATS, cell growth and invasion are suppressed. DATS reduces the expression of Notch1 and the related proteins VEGF, MMP-2, and MMP-9. By contrast, the expression levels of miR-34a, miR-143, miR-145, and miR-200b/c are increased. Among these miRNAs, miR-34a and miR-200b reduce Notch1 expression [96].

Circulating miRNAs in blood are targets for use as biomarkers (Table 2). Alkaline phosphatase (ALP) has been employed to monitor the progress of OS. ALP is related to event-free survival (EFS) and overall survival of patient with OS [97]. miR-195-5p, miR-199a-3p, miR-320a, and miR-374a-5p significantly increased in the plasma of OS patients comparing with healthy controls. The ROC curve for the four miRNAs shows the areas under the curve (AUC) of 0.96. In addition, the four miRNAs decreased after surgery, and the expressions of miR-195-5p and miR-199a-3p are significantly upregulated in the patients with metastasis comparing without metastasis [98]. Serum miR-195 has also been reported as a candidate of biomarkers in another study [99]. The serum miR-195 expression of OS patients is significantly downregulated comparing with healthy controls. The low expression of serum miR-195 is related to poor prognosis. In addition, miR-let7A expression is suppressed in both tissue and blood samples from OS patients. Its high expression is correlated with a good prognosis [100]. Furthermore, miR-9, miR-21, miR-199a-3p, and miR-143 have been reported as other candidates [101,102].

There are several reports about OS exosomes (Figure 3). Troyer et al. analyzed the proteomic cargo in the exosomes from OS cells [105]. The OS exosome was found to contain immunosuppressive proteins such as TGF-β, α fetoprotein, and heat shock proteins. The OS exosome suppresses T cell proliferation and activation [105]. Exosomal miR-143 derived from MSCs inhibits the migration of OS cells [106]. Next, it has been reported that exosomes derived from highly malignant or drug-resistant cancers expand the malignancy or drug-resistance [107,108]. Compared with the exosomes of nonmetastatic OS cells, the exosomes of metastatic OS cell lines significantly promote migration and invasion in osteoblast cells. miR-675 is significantly highly expressed in the exosomes of metastatic OS cells. Exosomal miR-675 suppresses the expression of calneuron1 (CALN1), which is a migration-related gene [109]. The exosome of a doxorubicin-resistant OS cell was found to induce a resistant phenotype in original cells. The exosome contained significantly higher multidrug resistance 1 (MDR-1) mRNA and P-glycoprotein (P-gp) and promoted the invasive ability of the original cells [110]. Cancer-associated stromal fibroblasts (CAFs) exist surrounding cancer cells and are functionally and phenotypically different from normal fibroblasts [111]. The exosomes of CAFs promote the migration and invasion of OS. miR-1228 is highly expressed in cells and exosomes of CAFs. Exosomal miR-1228 is transferred to OS cells and targets the gene called suppressor of cancer cell invasion [112].

Several exosomal miRNAs have been reported as OS biomarkers (Table 2). miR-25-3p is upregulated in the exosomes derived from OS cell lines. The serum miR-25-3p increases with tumor progression in vivo. In addition, the expression level of miR-25-3p is related to a poor prognosis and decreases after the resection of OS tumors [103]. Xu et al. analyzed the expression of exosomal miRNAs in the serum of OS patients before chemotherapy [104]. They compared the exosomal miRNA expression levels between patients with poor chemotherapeutic responses and those with good responses. Exosomal miR-135b, miR-148a, and miR-27a, and miR-9 were upregulated, and exosomal miR-124, miR-133a, miR-199a-3p, and miR-385 were downregulated in poor responders. These exosomal miRNAs may become biomarkers to predict the response of neo-adjuvant chemotherapy [104].

The understanding of mechanisms in drug resistance and metastasis is essential to develop treatment in OS, thus further studies of miRNAs and exosomes in OS will provide the novel therapy for OS treatments. In addition, since there is no useful biomarker for the effect of chemotherapy and the early detection of recurrence and metastasis, circulating miRNAs and exosomal miRNAs in blood can be new promising biomarker for OS.

### 4.2. Chondrosarcoma

Chondrosarcoma (CS) is the second most common primary bone sarcoma after OS. The peak incidence occurs at 50 to 70 years old. Surgery is almost the only CS treatment because CS is very resistant to chemotherapy and radiotherapy [113,114,115]. Therefore, CS is one of the sarcomas that are especially targeted for new treatments. We have shown that miRNAs regulate angiogenesis, Src, the mechanistic target of rapamycin (mTOR) pathway, and SRY-related high mobility group box gene (SOX), which will be new targets for CS treatment.

CS is classified into three grades along the malignancy. Angiogenesis has been seen in tumor tissue to be associated with the upgrading of malignancy [116]. There are several reports about the regulation of VEGF and angiogenesis through the control of miRNAs in CS. VEGF-A especially regulates angiogenesis among the subtypes of VEGF. Chemokine C-C motif receptor 5 (CCL5) is an inflammatory chemokine that promotes angiogenesis in ovarian and breast cancer [117,118]. CCL5 promotes the expression of VEGF-A by suppressing the expression of miR-199a and miR-200b, which are direct suppressors of VEGF-A [119,120]. WNT1-inducible signaling pathway protein-3 (WISP-3) and amphiregulin also promote VEGF-A expression by suppressing miR-452 and miR-206 [121,122]. In addition, hypoxia promotes angiogenesis in CS. Hypoxia increases miR-181a expression in CS. miR-181a directly targets regulator of G-protein signaling 16 (RGS16) and CXC chemokine receptor 4 (CXCR4), which promotes VEGF expression and angiogenesis [123]. Furthermore, VEGF-C is highly expressed in CS tissue, which especially promotes lymphangiogenesis. Basic fibroblast growth factor (bFGF) promotes VEGF-C expression in CS cells by suppressing miR-381 expression [124]. Similarly, brain-derived neurotrophic factor (BDNF) promotes VEGF-C expression by suppressing miR-624-3p via the mTOR signaling pathway [125]. In addition, leptin and adiponectin, which are secreted by adipocytes, promote VEGF-C expression in CS by suppressing miR-27b expression [126,127].

The c-Src pathway plays a role in cell survival, angiogenesis, and migration. The activation of c-Src has been identified in many types of cancers [128]. As described above, miR-452, miR-206, and miR-381 regulate VEGF via the c-Src pathway [121,122,124]. In addition, miR-23b promotes the sanitization of cisplatin in CS cells [129]. miR-141 and miR-101 inhibit the metastatic ability of CS [130,131]. All of them target the c-Src pathway. On the other hand, mTOR plays a key role in, not only the regulation of cell growth and homeostasis, but also pathological conditions, such as type 2 diabetes and cancer [132]. miR-100 directly suppresses mTOR expression and promotes the sensitization of cisplatin [129]. When the CS cells are treated with an mTOR inhibitor, the expression pattern is changed. miR-20a, miR-125b, and miR-192 are upregulated, but miR-509-3p, miR-589, miR-490-3p, and miR-550 are downregulated [133].

SOX9 is an important transcription factor regulating chondrocyte development [134]. miR-494 and miR-154 directly suppress SOX9 [135,136]. A low expression of miR-494 is correlated with poor prognosis. miR-154 suppresses ETV5 and MMP-2, which are downstream of SOX9. SOX4 also regulates chondrogenesis. miR-129-5p suppresses proliferation and migration and promotes apoptosis in CS cells by inhibiting SOX4 [137].

Little is known about the high resistance of chemotherapy and radiotherapy in CS. In addition, it is well known that angiogenesis is one of the factors characterizing CS. Thus, understanding the contribution of exosome for angiogenesis in CS might lead to the development of novel treatment against a resistance of chemotherapy and radiotherapy in CS.

### 4.3. Ewing’s Sarcoma

Ewing’s sarcoma (EWS) is a small round sarcoma that is the second most common sarcoma of bone in children. Approximately 80% of EWSs have chromosomal translocation t(11;22)(q24;q12), which results in the production of EWS-FLI1 oncoprotein [138]. Two-thirds of ES patients are cured, but the outcome for patients with disseminated or early relapse remains dismal [139]. EWS-FLI1 functions as a transcriptional factor that regulates crucial processes such as cell growth, apoptosis, and differentiation in EWS. When hMSCs overexpress EWS-FLI1, the hMSCs indicated similar gene expression profiles as EWS cells [140]. On the other hand, downregulation of EWS-FLI1 in EWS cells results in the similar gene expression of mesenchymal progenitor cells [141]. miRNAs related to EWS-FLI1 are summarized below.

When EWS-FLI1 is suppressed in EWS cells, the expression levels of miR-100, miR-125b, miR-22, miR-221/222, miR-27a, and miR-29a are suppressed. All of these miRNAs regulate the insulin-like growth factor (IGF) pathway and promote EWS cell growth [142]. miR-145 is also related to EWS-FLI1 expression. miR-145 suppresses EWS cell growth [143]. EWS-FLI1 protein upregulates EYA3 protein by suppressing miR-708. Suppressing EYA3 improves drug susceptibility [144].

CD99 is constantly present in EWS tissue, and CD99 is strongly related to EWS malignancy and expression of EWS-FLI1 [145]. CD99 expression is regulated by miR-34a via Notch signaling and prevents EWS differentiation. In addition, CD99 is also expressed on the exosomes derived from EWS cells [146]. In addition, an analysis of 49 primary EWS tumors revealed that miR-34a is related to EWS progression. The low expression of miR-34a predicts poor event-free and overall survival [147].

IGF/Akt/mTOR pathway is one of the key targets EWS treatment. The IGF receptor 1 (IGF-1R) and mTOR inhibitors prevent tumor growth and improve survival rate of EWS in xenograft model [148]. The study for regulation of miRNAs and exosomes in this pathway will provide the further development of the new treatment in EWS.

### 4.4. Rhabdomyosarcoma

Rhabdomyosarcomas (RMSs) are a heterogeneous group of malignant tumors and are the most frequent soft tissue sarcoma in children [149]. There are two major subtypes—embryonal RMS (ERMS) and alveolar RMS (ARMS). ERMS shows a histology similar to embryonic skeletal muscle, which develops preferentially in children under 10 years old. ARMS develops in adolescents and young adults and has specific chromosomal translocations, namely, t(2;13)(q35;q14) and t(1;13)(p36;q14). These translocations result in PAX3-FOXO1 and PAX7-FOXO1, which are found in approximately 70% of ARMSs [150]. RMS is inhibited from muscle differentiation [151]. We have shown that miRNAs are related to the fusion gene and to myogenic differentiation.

Several studies have shown that myogenesis is regulated by some growth factors, such as IGF and TGF-β [152]. Among them, TGF-β1 inhibits skeletal muscle differentiation via miR-24 regulation [153]. In addition, miR-450-5p regulates TGF-β1 expression by suppressing ENOX2 and PAX9 expression [154]. Furthermore, overexpression of miR-1 and miR-206 in RMS cells restores differentiation and blocks the tumorigenic phenotype in vivo [155]. Ghayad et al. reported an exosome of RMS [156]. They used five RMS cell lines and compared the expression levels of exosomal miRNAs. Some miRNAs, such as miR-1246, were found to be upregulated in the exosomes compared with donor cells. In addition, RMS exosomes affect normal fibroblasts and endothelial cells, which results in improved migration, invasion, and angiogenesis. The inhibition of muscle differentiation plays a key role in the pathology of RMS, thus further understanding of the regulatory miRNAs in muscle differentiation is required for further understanding the molecular mechanisms of RMS development.

### 4.5. Liposarcoma

Liposarcoma (LPS) is a common subtype of soft tissue sarcoma. The incidence rate of LPS is 20%, and LPS is classified into four categories: Well-differentiated liposarcoma (WDLPS), dedifferentiated liposarcoma (DDLPS), myxoid liposarcoma (MLPS), and pleomorphic liposarcoma (PLPS). WDLPS and DDLPS share similar chromosomal amplification, which is 12q14–15 involving MDM2 and CDK4 genes [2]. MDM2 inhibits the activity of P53 and decreases apoptosis. CDK4 phosphorylates Rb-gene products to promote the cell cycle. In addition, because many DDLPSs occur focally in WDLPS tissues, DDLPS is recognized to progress from DDLPS. MLPS has a specific chromosomal translocation t(12; 16)(q13; p11), which results in FUS-CHOP gene fusion, presenting in >95% of cases [157]. The success of chemotherapy and radiotherapy for MLPS is clearly better than that for the other subtypes of LPS, and the sites of metastasis of MLPS, such as bone and retroperitoneum, are different [158]. These facts indicate that the lineage of MLPS is different from WDLPS and DDLPS.

miR-155 is one of the most studied miRNAs in LPS, which is upregulated in LPS tissue and promotes tumor formation. Comparing the expression of miRNAs in the formalin-fixed paraffin-embedded (FFPE) tissue samples between LPSs and normal adipose tissues, miR-155 was upregulated in all LPS subtypes, including WDLPS, DDLPS, MLPS, and PLPS [159]. Kapodistrias et al. [160] also compared 83 FFPE tissue samples between LPSs with lipomas. miR-155 and miR-21 were upregulated, whereas miR-143 and miR-145 were downregulated. In particular, the high expression of miR-155 indicated a correlation with poor prognosis, while the other miRNAs did not show a relationship with LPS prognosis. miR-155 targeted casein kinase 1α (CK1α), which is related to the Wnt pathway. miR-155 knockdown inhibited DDLPS growth in vitro and in vivo.

miR-26a-2 is a gene at 12q14 that is adjacent to MDM2 and is coamplified with MDM2. miR-26a-2 amplification is significantly high not only in WDLPS and DDLPS but also in MLPS. High miR-26a-2 expression is correlated with a poor prognosis in WDLPS, DDLPS, and MLPS. The overexpression of miR-26a-2 in DDLPS cells induces cell growth and the inhibition of adipose differentiation [161]. In addition, miR-2a-2 promotes apoptosis by targeting RCBTB1 and homeobox protein A5 (HOXA5). RCBTB1 is a regulator of the DNA damage and repair pathway and apoptosis, and HOXA5 shows a correlation with adipocyte differentiation and fat metabolism [162,163].

miR-25-3p and miR-92a-3p are upregulated in the exosomes of several LPS cell lines. These exosomal miRNAs are transferred to macrophages and activate IL-6 secretion. Secreted IL-6 is transferred to LPS cells and upregulates LPS aggressiveness [164].

It is still unknown what develops WDLPS into DDLPS, which might be partially regulated by miRNAs. Gits et al. examined miRNA expressions in the tissue samples of LPS which included all subtypes, lipomas, and normal fat [165]. They identified that the miRNA profile changes significantly during the dedifferentiation of WDLPS. From this point of view, further functional studies of miRNA and exosomal miRNA are needed in order to understand the underlying mechanisms of LPS.

## 5. Conclusions

In this review, we summarized current research on the miRNAs and exosomes of sarcomas. Recent advances in miRNAs and exosomes have led to the understanding of entirely new mechanisms of sarcoma development and progression. However, more experiments on miRNAs and exosomes in vitro and in vivo are required to confirm the relevant target genes.

Chromosomal aberrations play a key role in OS tumorigenesis and lead to subsequent amplification and deficiency of specific gene regions. Among them, the 6p12–p21 region is one of the most frequently amplified, which includes Runx2. Runx2 is essential for osteoblastic differentiation and skeletal morphogenesis, and high expression of Runx2 is positively related to a poor prognosis of OS. It is noteworthy that p53 suppresses Runx2 expression in OS cells via miR-34 regulation. In this manner, the studies of miRNAs related to the regulation of these gene regions will elucidate the pathology of OS and open up possibilities for new therapeutic agents. In addition, the elucidation of metastasis and drug resistance is one of the most important subjects of OS research. We have shown the exosomal functions of the tumor microenvironments and applications as drug-resistance biomarkers of OS. In other cancers, the treatments targeting exosomes secreted by cancers have been reported, which are inhibition of exosomal production, targeting of circulating exosomes, and inhibition of receiving exosomes [166,167,168]. Such treatments will be greatly applicable in the OS field.

CS is the second most common bone tumor and shows strong drug resistance and radiotherapy resistance. The expression of VEGF and angiogenesis are related to poor prognosis of CS. VEGF is a promising target for the CS drug development. However, there is a left target for miRNA and exosome research that is still needed in the CS field. For instance, there are no miRNA studies of IDH1/2 mutation, which plays a key role in the diagnosis and pathogenesis of CS. Further studies of miRNA and exosomes are expected in the field of CS research.

EWS and RMS have a selection of specific fusion genes. The expression of these fusion proteins is related to tumorigenesis and inhibition of cancer cell development. However, the functions of these fusion genes have not been elucidated sufficiently. We have described reports about miRNA and exosomal functions related to these fusion genes, but it is desirable to further analyze miRNAs and exosomes.

In conclusion, it is essential to understand the precise molecular mechanisms of miRNA-mediated and exosomal miRNA-mediated sarcoma progression. This knowledge would lead to the development of novel diagnostics and treatments for sarcoma by miRNAs and exosomal miRNAs. It has been proposed that there are several methods for exosome-targeted therapy to treat cancer [169]. Thus, further investigation of miRNAs and exosomal miRNAs in sarcomas will contribute to the cure of sarcomas.

## Figures and Tables

**Figure 1 cancers-11-00428-f001:**
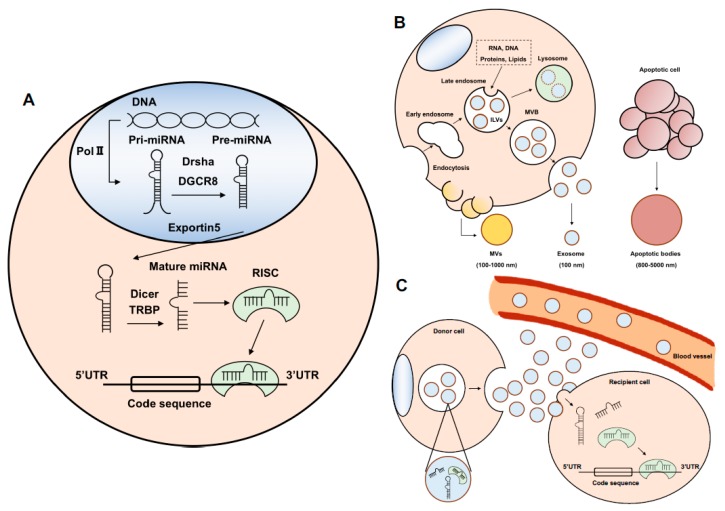
The biogenesis of miRNA and exosomes. (**A**) The biogenesis of miRNA. Pri-miRNA is transcribed by RNA polymerase II (Pol II). Pri-miRNA is cleaved by the microprocessor (Drsha and DGCR8) to pre-miRNA. Pre-miRNA is in the form of a hairpin. Pre-miRNA is exported from the nucleus to the cytoplasm by Exportin 5. Pre-miRNA binds to Dicer and trans-activation-responsive RNA-binding protein (TRBP) and is cut to a single RNA called mature miRNA. The miRNA binds to Argonaute (AGO) protein. This complex of miRNA and AGO protein names the RNA-induced silencing complex (RISC). The mature miRNA binds to the targeting of miRNAs with complementary sites and results in translational repression or mRNA degradation. (**B**) The biogenesis of exosomes. Exosomes are produced through the pathway of endocytosis. The origin of exosomes is the intraluminal vesicles (ILVs) in late endosomes, which develop into multivesicular bodies (MVBs) or fuse with lysosomes. At this stage, RNA, including miRNA, proteins, and lipids are encapsulated into the ILVs. These vesicles are released into the extracellular space and are called exosomes. By contrast, microvesicles (MVs) are produced from the cell membrane independently of the endocytosis pathway. Apoptotic bodies are formed from apoptotic cells. (**C**) The releasing exosomal miRNA. The exosome is transferred from the donor cells to recipient cells. The exosomal miRNAs are released into recipient cells and regulate transcription. On the other hand, some exosomal miRNAs are released into bodily fluids, such as blood, and carried to distant organs.

**Figure 2 cancers-11-00428-f002:**
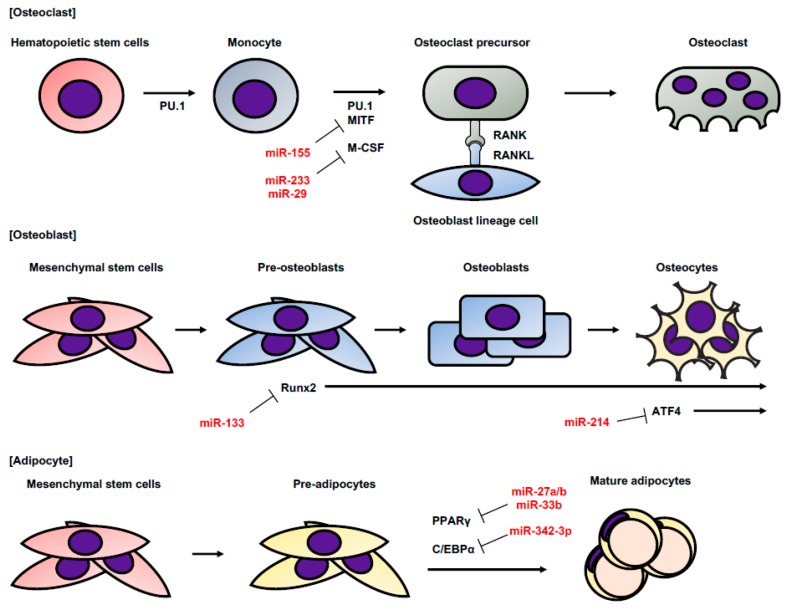
The differentiations of osteoclasts, osteoblasts, and adipocytes. This figure shows the order of normal differentiation of osteoclasts, osteoclasts, and adipocytes. The genes related to their differentiation at each stage and the regulatory miRNAs are also shown. This figure is drawn with reference to References [12,18].

**Figure 3 cancers-11-00428-f003:**
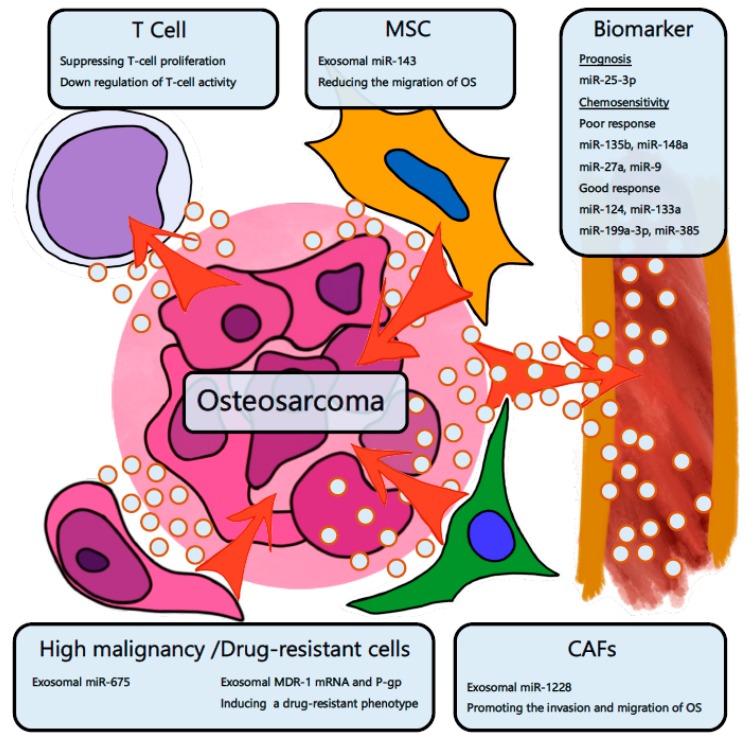
The exosome in the tumor microenvironment of osteosarcoma. OS cells suppress the activity of T cells by transferring exosomes. Mesenchymal stem cells (MSCs) transfer exosomal miR-143 to OS cells and suppress OS aggressiveness. High malignancy and drug-resistant cells of OS transfer exosomal miR-675, exosomal MDR-1, and P-gp mRNA and expand the malignancy. Cancer-associated stromal fibroblasts (CAFs) transfer miR-1228 to OS cells and support OS invasion. Several exosomal miRNAs can be used as biomarkers for OS prognosis and chemosensitivity.

**Table 1 cancers-11-00428-t001:** The gene amplification regions of Osteosarcoma and relating genes and miRNAs.

**Amplification Gene Region**	**Located Genes**	**Relating miRNAs**	**Functions**	**Reference**
6 p12–p21	Runx2	miR-34c, miR-338-3p, miR-23a, miR-203	Directly suppressing Runx2 expression	[50,51,52,53]
	VEGF, VEGFR	miR-134, miR-145, miR-20b	Suppressing VEGF and VEGFR expressions	[61,62,63]
		miR-337-5p	Activating VEGF, ERBB, and MAPK pathway	[64]
	CDC5L			
17p11	MAPK7, MAP2K4	miR-143, miR-125b	Directly suppressing MAPK7 expression	[67,68]
12 q13–q14	CDK4, MDM2	miR-506-3p	Suppressing CDK4 and MMP9 expression via targeting RABD3D	[69]
8q21–24	Myc	miR-33b, miR-449c, miR-135b	Directly suppressing c-Myc expression	[70,71,72]
		miR-214	Promoting c-Myc expression via Wnt signaling	[73]
		miR-107, miR-137	Suppressing c-Myc expression via Wnt signaling	[74,75]
**Loss of Gene Region**	**Located Genes**	**Relating miRNAs**	**Functions**	**Reference**
17p13	TP53	miR-381	Promoting p53 expression	[76]
		miR-373	Suppressing p53 expression	[77]
		miR-1281	P53 regulates miR-1281 expression.	[78]
		miR-527, miR-665	⊿Np63α suppresses miR-527 and miR-665 expressions.	[79]
13q14	RB1	miR-142	Promoting Rb expression	[80]

**Table 2 cancers-11-00428-t002:** The promising miRNAs as biomarkers of Osteosarcoma (OS).

**miRNAs**	**Direction of miRNA Expression in OS Samples**	**The Types and Sizes of Samples**	**Role**	**Source**	**Findings**	**Reference**
miR-195-5p, miR-199a-3p, miR-320a, miR-374a-5p	↑	90 pre-operative OS pts. vs. 90 healthy ctrls. 90 pre-operative OS pts. vs. 50 post-operative pts.	Diagnosis	Plasma	Distinguishing between OS patients and healthy controls, with AUC of 0.96 The four miRNAs decrease after operation.	[98]
miR-195	↓	166 OS pts. vs. 60 healthy ctrls.	Diagnosis	Serum	Distinguishing between OS patients and healthy controls, with AUC of 0.892 Relating with prognosis of OS	[99]
miR-Let7A	↓	OS tissues vs. pericarcinomatous tissues (39 OS pts.) 39 OS pts. vs. 19 healthy ctrls.	Diagnosis	Tissue, Blood	Distinguishing between OS patients and healthy controls by blood miR-Let7A, with AUC of 0.90 The expression of blood miR-Let7A relates with prognosis of OS.	[100]
miR-9	↑	118 OS pts. vs. 60 healthy ctrls.	Diagnosis	Serum	Distinguishing between OS patients and healthy controls Relating with prognosis of OS.	[101]
miR-21	↑	40 OS pts. vs. 40 healthy ctrls.	Diagnosis	Plasma	Distinguishing between OS patients and healthy controls by combined the three miRNAs, with AUC of 0.95	[102]
miR-199a-3p, miR-143	↓					
miR-25-3p	↑	Discovery set 10 OS pts. vs. 10 healthy ctrls., 10 OS pts. vs. 10 non-OS pts., 2 OS pts. (preoperative state) vs. 2 OS pts. (postoperative state), 7 culture media from OS cell lines 7 exosomes from OS cell lines Serum miRNA of OS xenograft Validation cohort 14 OS pts. vs. 14 non-OS pts. vs. 10 healthy ctrls.	Diagnosis	Serum	Distinguishing between OS patients and healthy controls, with AUC of 0.87 Decreasing after operation. Relating with prognosis of OS	[103]
**miRNAs**	**Direction of miRNA Expression in Good Responders**	**The Types and Sizes of Samples**	**Role**	**Source**	**Findings**	**Reference**
miR-135b, miR-148a, miR-27a, miR-9	↑	25 OS pts. with good response vs. 28 OS pts. with poor response vs. 31 healthy ctrls.	chemotherapeutic response	Exosomes from sera	Distinguishing between good responders and poor responders, with AUC from 0.865 to 0.938	[104]
miR-124, miR-133a, miR-199a-3p, miR-385	↓					

Patients (pts), controls (ctrls).

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
