# Peer review of "An Insight into the Roles of MicroRNAs and Exosomes in Sarcoma"

_cancers, 2019, doi:10.3390/cancers11030428_

Round 1

Reviewer 1 Report

It is comprehensive in its coverage of the subject. It could certainly be decreased in size by 20-30% without sacrificing quality of content; there are numerous redundancies in the text.

Author Response

We are grateful to all of reviewers for the critical comments and insightful suggestions that have helped us to improve our paper considerably. As indicated in the responses that follow, we have taken all of these comments and suggestions into account in the revised version of our manuscript.

Query No.1).  It is comprehensive in its coverage of the subject. It could certainly be decreased in size by 20-30% without sacrificing quality of content; there are numerous redundancies in the text.

Answer No.1) According to the Reviewer’s suggestion, we have checked throughout the manuscript and deleted redundant sentences.

List of Changes Made

1. Page 1, line21: “microRNA” was changed to “miRNA”.

2. Page 1, lines 38 to 39: “MicroRNA (miRNA) is a type of noncoding RNA (ncRNA) that consists of 17-25 base pairs of short ncRNA and regulates gene expression mainly at the transcriptional level.” was changed to “MicroRNA (miRNA) is a type of noncoding RNA (ncRNA) that consists of 17-25 base pairs of short ncRNA and regulates gene expression mainly by suppressing translation or enhancing mRNA degradation via targeting the 3’-UTR, which is post transcriptional regulation of gene expression.”.

3. Page 2, figure1: figure 1B was changed, which is added the figure of fusing with lysosomes.

4. Page2, line 51: “that consists of 17-25 base pairs of short noncoding RNA” and “miRNA regulates gene expression at the transcriptional level by binding to the target miRNA.” were deleted.

5. Page 2, lines 59 to 60: “RISC binds to targeting mRNA and regulates translation.” was changed to “The mature miRNA reads to the targeting of miRNAs with complementary sites and results in translational repression or mRNA degradation.”.

6. Page 2, lines 62 to 63: “which develop into multivesicular bodies (MVBs) or fuse with lysosomes.” And “including miRNA, DNA, proteins and lipids” were added.

7. Page 4, line 112: “In addition, we will show the utility of exosomal miRNAs as biomarkers for diagnosis.” was deleted.

8. Pages 5 to 6, table 1: the functions or targets of miRNAs were added.

9. Page 7, lines 228 to 235: the functional explanations of circulating miRNAs in Osteosarcoma were added, which was “miR-195-5p, miR-199a-3p, miR-320a and miR-374a-5p significantly increased in the plasma of OS patients comparing with healthy controls. The ROC curve for the four miRNAs shows the areas under the curve (AUC) of 0.96. In addition, the four miRNAs decreased after surgery, and the expressions of miR-195-5p and miR-199a-3p are significantly upregulated in the patients with metastasis comparing without metastasis [96]. Serum miR-195 has also been reported as a candidate of biomarkers in another study [97]. The serum miR-195 expression of OS patients is significantly downregulated comparing with healthy controls. The low expression of serum miR-195 is related with poor prognosis. ”. 

10. Page 8, lines 266 to 270: “The clarification of mechanisms in drug resistance and metastasis is essential to develop treatment in OS. Accumulating studies of miRNAs and exosomes in OS will provide the new way forward for OS treatments. In addition, since there is no useful biomarker for the effect of chemotherapy and the early detection of recurrence and metastasis, circulating miRNAs and exosomal miRNAs in blood can be new promising biomarker for OS.” was added.

11. Pages 9 to 10: Table 2 was added.

12. Page 11, lines 325 to 328: “Little is known about the high resistance to chemotherapy and radiotherapy in CS. Even though CS and OS are the tumors arising from similar cell lineage, it is of interest to note about the difference of sensitivity for treatment between them. Angiogenesis is one of the factors characterizing CS, which results in maintaining the fugue tumor size of CS and weakening the effect of treatment.”.

13. Page 12, lines 351 to 354: “The clarification of the mechanisms underlying EWS-FLI1 in EWS must lead to improve the treatment. IGF/Akt/mTOR pathway is one of the key target for treatment of EWS. The treatment of IGF receptor 1 (IGF-1R) and mTOR inhibitors prevent tumor growth and improve survival rate of EWS xenografts [148]. The study for regulation of miRNAs and exosomes with this pathway will proceed with development of the new treatment in EWS.” was added.

14. Page 12, lines 373 to 376: “The inhibition of muscle differentiation plays a key role in the pathology of RMS and the regulatory miRNAs are required further investigation.” was added.

15. Page 13, lines 411 to 416: “It is not clear what develops WDLPS into DDLPS, which must be partially regulated by miRNAs. Gits et al. examined miRNA expressions in the tissue samples of LPS which included all subtypes, lipomas and normal fat [164]. They indicated that the miRNA profile changes significantly during the dedifferentiation of WDLPS. Further studies of miRNA and exosomal miRNA are needed in order to understand the underlying mechanisms of LPS.” was added.

14. Page 17, line 573: the mistake of reference 40 was corrected.

15. Page 18, line 597: the mistake of reference 47 was corrected.

16. Page 18, line 604: the mistake of reference 49 was corrected.

17. Page 19, line 625: the mistake of reference 57 was corrected.

18. Page 19, line 628: the mistake of reference 58 was corrected.

19. Page 22, line 675 : the mistake of reference 74 was corrected.

20. Page 26, line 843 : the mistake of reference 133 was corrected.

Reviewer 2 Report

This is a rather comprehensive review of the role of miRNAs and exosomes in Sarcoma by Dr. Ochiya’s group. This is a research topic they have been touched upon over the years, and this review provides an update of recent development in the studies of miRNAs and exosomes relating to Sarcoma.

There are some minor concerns that need to be addressed.

After microRNA is abbreviated as miRNA, the form of microRNA is still used in the writing (line 21, page 1).

It is well accepted that miRNAs regulate gene expression mainly by suppressing translation or enhancing mRNA degradation via targeting the 3’-UTR, which is post transcriptional regulation of gene expression. The statement that miRNA regulates gene expression mainly at the transcriptional level (line 37, page 1; line 52, 68, page 2) is incorrect.

Figure 1B should be modified to indicate that late endosomes may be developed into MVBs or fused with lysosomes. The current version may lead to misunderstanding that all endosomes will be developed into MVBs.

While the manuscript was fairly drafted, a professional English editing may improve the quality of the manuscript.

Author Response

We are grateful to all of reviewers for the critical comments and insightful suggestions that have helped us to improve our paper considerably. As indicated in the responses that follow, we have taken all of these comments and suggestions into account in the revised version of our manuscript.

Query No.1). After microRNA is abbreviated as miRNA, the form of microRNA is still used in the writing (line 21, page 1).

Answer No.1) We apologize for making this mistake. We corrected it as shown in line 21 at page 1. In addition, we carefully checked throughout the manuscript for collecting this.

Query No.2). It is well accepted that miRNAs regulate gene expression mainly by suppressing translation or enhancing mRNA degradation via targeting the 3’-UTR, which is post transcriptional regulation of gene expression. The statement that miRNA regulates gene expression mainly at the transcriptional level (line 37, page 1; line 52, 68, page 2) is incorrect. 

Answer No.2) We apologized the mistake of this. As you can find in the revised manuscript, we corrected the sentences as shown in line 38 at page 1 and in page2 at line59.

Query No.3). Figure 1B should be modified to indicate that late endosomes may be developed into MVBs or fused with lysosomes. The current version may lead to misunderstanding that all endosomes will be developed into MVBs. 

Answer No.3) Thank you for your suggestion. We corrected the figure and legend of the figure1B as you can find at line 62-63 in page2.

Query No.4). While the manuscript was fairly drafted, a professional English editing may improve the quality of the manuscript.

Answer No.4) Our manuscript was corrected by a professional English editing, however, as suggested from Reviewer, we have carefully checked our manuscript again. Thank you for your kind suggestion.

List of Changes Made

1. Page 1, line21: “microRNA” was changed to “miRNA”.

2. Page 1, lines 38 to 39: “MicroRNA (miRNA) is a type of noncoding RNA (ncRNA) that consists of 17-25 base pairs of short ncRNA and regulates gene expression mainly at the transcriptional level.” was changed to “MicroRNA (miRNA) is a type of noncoding RNA (ncRNA) that consists of 17-25 base pairs of short ncRNA and regulates gene expression mainly by suppressing translation or enhancing mRNA degradation via targeting the 3’-UTR, which is post transcriptional regulation of gene expression.”.

3. Page 2, figure1: figure 1B was changed, which is added the figure of fusing with lysosomes.

4. Page2, line 51: “that consists of 17-25 base pairs of short noncoding RNA” and “miRNA regulates gene expression at the transcriptional level by binding to the target miRNA.” were deleted.

5. Page 2, lines 59 to 60: “RISC binds to targeting mRNA and regulates translation.” was changed to “The mature miRNA reads to the targeting of miRNAs with complementary sites and results in translational repression or mRNA degradation.”.

6. Page 2, lines 62 to 63: “which develop into multivesicular bodies (MVBs) or fuse with lysosomes.” And “including miRNA, DNA, proteins and lipids” were added.

7. Page 4, line 112: “In addition, we will show the utility of exosomal miRNAs as biomarkers for diagnosis.” was deleted.

8. Pages 5 to 6, table 1: the functions or targets of miRNAs were added.

9. Page 7, lines 228 to 235: the functional explanations of circulating miRNAs in Osteosarcoma were added, which was “miR-195-5p, miR-199a-3p, miR-320a and miR-374a-5p significantly increased in the plasma of OS patients comparing with healthy controls. The ROC curve for the four miRNAs shows the areas under the curve (AUC) of 0.96. In addition, the four miRNAs decreased after surgery, and the expressions of miR-195-5p and miR-199a-3p are significantly upregulated in the patients with metastasis comparing without metastasis [96]. Serum miR-195 has also been reported as a candidate of biomarkers in another study [97]. The serum miR-195 expression of OS patients is significantly downregulated comparing with healthy controls. The low expression of serum miR-195 is related with poor prognosis. ”. 

10. Page 8, lines 266 to 270: “The clarification of mechanisms in drug resistance and metastasis is essential to develop treatment in OS. Accumulating studies of miRNAs and exosomes in OS will provide the new way forward for OS treatments. In addition, since there is no useful biomarker for the effect of chemotherapy and the early detection of recurrence and metastasis, circulating miRNAs and exosomal miRNAs in blood can be new promising biomarker for OS.” was added.

11. Pages 9 to 10: Table 2 was added.

12. Page 11, lines 325 to 328: “Little is known about the high resistance to chemotherapy and radiotherapy in CS. Even though CS and OS are the tumors arising from similar cell lineage, it is of interest to note about the difference of sensitivity for treatment between them. Angiogenesis is one of the factors characterizing CS, which results in maintaining the fugue tumor size of CS and weakening the effect of treatment.”.

13. Page 12, lines 351 to 354: “The clarification of the mechanisms underlying EWS-FLI1 in EWS must lead to improve the treatment. IGF/Akt/mTOR pathway is one of the key target for treatment of EWS. The treatment of IGF receptor 1 (IGF-1R) and mTOR inhibitors prevent tumor growth and improve survival rate of EWS xenografts [148]. The study for regulation of miRNAs and exosomes with this pathway will proceed with development of the new treatment in EWS.” was added.

14. Page 12, lines 373 to 376: “The inhibition of muscle differentiation plays a key role in the pathology of RMS and the regulatory miRNAs are required further investigation.” was added.

15. Page 13, lines 411 to 416: “It is not clear what develops WDLPS into DDLPS, which must be partially regulated by miRNAs. Gits et al. examined miRNA expressions in the tissue samples of LPS which included all subtypes, lipomas and normal fat [164]. They indicated that the miRNA profile changes significantly during the dedifferentiation of WDLPS. Further studies of miRNA and exosomal miRNA are needed in order to understand the underlying mechanisms of LPS.” was added.

14. Page 17, line 573: the mistake of reference 40 was corrected.

15. Page 18, line 597: the mistake of reference 47 was corrected.

16. Page 18, line 604: the mistake of reference 49 was corrected.

17. Page 19, line 625: the mistake of reference 57 was corrected.

18. Page 19, line 628: the mistake of reference 58 was corrected.

19. Page 22, line 675: the mistake of reference 74 was corrected.

20. Page 26, line 843: the mistake of reference 133 was corrected.

Reviewer 3 Report

cancers-455235:

The authors summarized the roles of miRNAs an exosomes in sarcoma and suggested their function in cancer treatment.

I have some comments on this manuscript.

Major points:

1. Although the authors tried to explain their story, the Reviewer wonder that there are too many references before 2000 or 2010 also which the authors used for their statement.

It would be better if the authors can update the references timely.

2. Several sentences are missing references.

For example:  Line 37, Page 1 of 23.

3. Similarly, the authors also should cite references for their Figures.

4. Part 4 (4.1 – 4.5): The authors should provide more their opinion and insightful comments in each part, not only summarizing the key findings.

5. Table 1 is too short, it should be extended more. The authors should also shortly provide the related mechanisms or functions in Table 1, hence, the Readers will be easier to follow and overview (not just make a list).

6. The authors mentioned several times in the manuscript that exosomes and exosomal miRNAs can be used as cancer biomarkers. The Reviewer would like to suggest the authors to prepare one Table about this statement also.

Minor points:

1. The authors should carefully follow journal reference style.

For example: Ref 74.

Author Response

We are grateful to all of reviewers for the critical comments and insightful suggestions that have helped us to improve our paper considerably. As indicated in the responses that follow, we have taken all of these comments and suggestions into account in the revised version of our manuscript.

Major points:

Query No.1). Although the authors tried to explain their story, the Reviewer wonder that there are too many references before 2000 or 2010 also which the authors used for their statement. It would be better if the authors can update the references timely.

Answer No.1) Thank you for your indication. In the paragraph of “2. MicroRNA”, we would like to introduce the basic knowledge about the miRNAs in cancer development including sarcomas, thus we tried to introduce the earlier works which were shown the roles of miRNAs in sarcomas. On the other hands, in the paragraph of “3. Exosome and exosomal miRNA”, we have chosen current manuscript, especially after 2010 when the exosomal miRNAs were proved as mediator of cell-cell communications.[MOU1] 

Query No.2). Several sentences are missing references. For example:  Line 37, Page 1 of 23.

Answer No.2) We apologize for making these mistakes. We have carefully checked and added in the sentences appropriately.

Query No.3). Similarly, the authors also should cite references for their Figures.

Answer No.3) Thank you for your suggestions. We checked and added the references in the Figures appropriately.

Query No.4). Part 4 (4.1 – 4.5): The authors should provide more their opinion and insightful comments in each part, not only summarizing the key findings.

Answer No.4) Thank you for your insightful comments. We added the comments into each part. You can find these in the revised manuscript shown in red colour.

Query No.5). Table 1 is too short, it should be extended more. The authors should also shortly provide the related mechanisms or functions in Table 1, hence, the Readers will be easier to follow and overview (not just make a list).

Answer No.5) According to the Reviewer’s insightful comment, we added the functions of each miRNA in the Table1.

Query No.6). The authors mentioned several times in the manuscript that exosomes and exosomal miRNAs can be used as cancer biomarkers. The Reviewer would like to suggest the authors to prepare one Table about this statement also.

Answer No.6) Thank you for your suggestions. We made the table of miRNAs and exosomal miRNAs as osteosarcoma biomarkers (Table2).

Query No.7). 1. The authors should carefully follow journal reference style.

For example: Ref 74.

Answer No.7) We apologize for making these mistakes. We carefully confirmed each reference and corrected.

List of Changes Made

1. Page 1, line21: “microRNA” was changed to “miRNA”.

2. Page 1, lines 38 to 39: “MicroRNA (miRNA) is a type of noncoding RNA (ncRNA) that consists of 17-25 base pairs of short ncRNA and regulates gene expression mainly at the transcriptional level.” was changed to “MicroRNA (miRNA) is a type of noncoding RNA (ncRNA) that consists of 17-25 base pairs of short ncRNA and regulates gene expression mainly by suppressing translation or enhancing mRNA degradation via targeting the 3’-UTR, which is post transcriptional regulation of gene expression.”.

3. Page 2, figure1: figure 1B was changed, which is added the figure of fusing with lysosomes.

4. Page2, line 51: “that consists of 17-25 base pairs of short noncoding RNA” and “miRNA regulates gene expression at the transcriptional level by binding to the target miRNA.” were deleted.

5. Page 2, lines 59 to 60: “RISC binds to targeting mRNA and regulates translation.” was changed to “The mature miRNA reads to the targeting of miRNAs with complementary sites and results in translational repression or mRNA degradation.”.

6. Page 2, lines 62 to 63: “which develop into multivesicular bodies (MVBs) or fuse with lysosomes.” And “including miRNA, DNA, proteins and lipids” were added.

7. Page 4, line 112: “In addition, we will show the utility of exosomal miRNAs as biomarkers for diagnosis.” was deleted.

8. Pages 5 to 6, table 1: the functions or targets of miRNAs were added.

9. Page 7, lines 228 to 235: the functional explanations of circulating miRNAs in Osteosarcoma were added, which was “miR-195-5p, miR-199a-3p, miR-320a and miR-374a-5p significantly increased in the plasma of OS patients comparing with healthy controls. The ROC curve for the four miRNAs shows the areas under the curve (AUC) of 0.96. In addition, the four miRNAs decreased after surgery, and the expressions of miR-195-5p and miR-199a-3p are significantly upregulated in the patients with metastasis comparing without metastasis [96]. Serum miR-195 has also been reported as a candidate of biomarkers in another study [97]. The serum miR-195 expression of OS patients is significantly downregulated comparing with healthy controls. The low expression of serum miR-195 is related with poor prognosis. ”. 

10. Page 8, lines 266 to 270: “The clarification of mechanisms in drug resistance and metastasis is essential to develop treatment in OS. Accumulating studies of miRNAs and exosomes in OS will provide the new way forward for OS treatments. In addition, since there is no useful biomarker for the effect of chemotherapy and the early detection of recurrence and metastasis, circulating miRNAs and exosomal miRNAs in blood can be new promising biomarker for OS.” was added.

11. Pages 9 to 10: Table 2 was added.

12. Page 11, lines 325 to 328: “Little is known about the high resistance to chemotherapy and radiotherapy in CS. Even though CS and OS are the tumors arising from similar cell lineage, it is of interest to note about the difference of sensitivity for treatment between them. Angiogenesis is one of the factors characterizing CS, which results in maintaining the fugue tumor size of CS and weakening the effect of treatment.”.

13. Page 12, lines 351 to 354: “The clarification of the mechanisms underlying EWS-FLI1 in EWS must lead to improve the treatment. IGF/Akt/mTOR pathway is one of the key target for treatment of EWS. The treatment of IGF receptor 1 (IGF-1R) and mTOR inhibitors prevent tumor growth and improve survival rate of EWS xenografts [148]. The study for regulation of miRNAs and exosomes with this pathway will proceed with development of the new treatment in EWS.” was added.

14. Page 12, lines 373 to 376: “The inhibition of muscle differentiation plays a key role in the pathology of RMS and the regulatory miRNAs are required further investigation.” was added.

15. Page 13, lines 411 to 416: “It is not clear what develops WDLPS into DDLPS, which must be partially regulated by miRNAs. Gits et al. examined miRNA expressions in the tissue samples of LPS which included all subtypes, lipomas and normal fat [164]. They indicated that the miRNA profile changes significantly during the dedifferentiation of WDLPS. Further studies of miRNA and exosomal miRNA are needed in order to understand the underlying mechanisms of LPS.” was added.

14. Page 17, line 573: the mistake of reference 40 was corrected.

15. Page 18, line 597: the mistake of reference 47 was corrected.

16. Page 18, line 604: the mistake of reference 49 was corrected.

17. Page 19, line 625: the mistake of reference 57 was corrected.

18. Page 19, line 628: the mistake of reference 58 was corrected.

19. Page 22, line 675 : the mistake of reference 74 was corrected.

20. Page 26, line 843 : the mistake of reference 133 was corrected.

Round 2

Reviewer 1 Report

The manuscript is now ready for publication.

Author Response

Response to the comments from Reviewer.

Reviewer 1

Query No.1). The manuscript is now ready for publication.

Answer No.1) Thank you very much for taking the time for reviewing our manuscript.

List of Changes Made

1. Page1, line 39: a new reference [3] was added.

2. Page1, line 42: a new reference [4] was added.

3. page14, line 454-481: the abbreviations part was added as follows; “6. Abbreviations

microRNA (miRNA), noncoding RNA (ncRNA), messenger RNA (mRNA), RNA polymerase II (Pol II), trans-activation-responsive RNA-binding protein (TRBP), argonaute (AGO), RNA-induced silencing complex (RISC), intraluminal vesicles (ILVs), multivesicular bodies (MVBs), microvesicles (MVs), Toll-like receptor (TLR), IL-1R-associated kinase 1 (IRAK1), TNFR-associated factor 6 (TRAF6), TAK1-associated binding protein 2 (TAB2), vascular cell adhesion molecule 1 (VCAM-1), microphthalmia-associated transcription factor (MITF), macrophage colony-stimulating factor (M-CSF), receptor activator of nuclear factor kappa-B (RANK), RANK ligand (RANKL), nuclear factor I/A (NFIA), mesenchymal stem cells (MSCs), Runt-related transcription factor 2 (Runx2), CCAAT-enhancer-binding proteins (C/EBPs), proliferator-activated receptor γ (PPARγ), extracellular vesicles (EVs), bone marrow-derived MSCs (BM-MSCs), colorectal cancer cells (CRC), breast cancer (BC), Osteosarcoma (OS), vascular endothelial growth factor (VEGF), VEGF receptor (VEGFR), microvessel density (MVD), hypoxia inducible factor 1 α (HIF1α), erythroblastic leukemia viral oncogene homolog (ERBB), mitogen-activated protein kinase (MAPK), depletion of cell division cycle 5-like (CDC5L), cyclin-dependent kinase 4 (CDK4), mouse double minute 2 homolog (MDM2), matrix metalloproteinase (MMP), tyrosine kinase (RTK), dickkopf-related protein 1 (DKK1), Li–Fraumeni syndrome (LFS), endoplasmic reticulum (ER), transforming growth factor-β (TGF-β), cancer stem cells (CSCs), diallyl trisulfide (DATS), alkaline phosphatase (ALP), event-free survival (EFS), areas under the curve (AUC), calneuron1 (CALN1), multidrug resistance 1 (MDR-1), P-glycoprotein (P-gp), cancer-associated stromal fibroblasts (CAFs), chondrosarcoma (CS), mechanistic target of rapamycin (mTOR), SRY-related high mobility group box gene (SOX), chemokine C-C motif receptor 5 (CCL5), WNT1-inducible signaling pathway protein-3 (WISP-3), G-protein signaling 16 (RGS16), CXC chemokine receptor 4 (CXCR4), basic fibroblast growth factor (bFGF), brain-derived neurotrophic factor (BDNF), Ewing’s sarcoma (EWS), insulin-like growth factor (IGF), IGF receptor 1 (IGF-1R), rhabdomyosarcomas (RMSs), embryonal RMS (ERMS), alveolar RMS (ARMS), liposarcoma (LPS), well-differentiated liposarcoma (WDLPS), dedifferentiated liposarcoma (DDLPS), myxoid liposarcoma (MLPS), pleomorphic liposarcoma (PLPS), casein kinase 1α (CK1α), homeobox protein A5 (HOXA5)”.

4. Page15, line 500,501: New reference was added as follows; [3], “Lin, S.; Gregory, R.I. MicroRNA biogenesis pathways in cancer. Nat. Rev. Cancer 2015, 15, 321–33.”.

45 Page15, line 502,503: New reference was added as follows; [4], “Bach, D.-H.; Hong, J.-Y.; Park, H.J.; Lee, S.K. The role of exosomes and miRNAs in drug-resistance of cancer cells. Int. J. cancer 2017, 141, 220–230.”.

Reviewer 3 Report

cancers-455235:

1. Introduction part:

Line 39, Page 1 of 28: The authors should cite PMID: 25998712

Line 41, Page 1 of 28: The authors should cite PMID: 28240776

2. The authors should prepare Abbreviations part (with order in alphabet) for this manuscript, hence, the Readers can be easier to follow also.

Author Response

Response to the comments from Reviewer.

Reviewer 3

Query No.1)

1. Introduction part:

Line 39, Page 1 of 28: The authors should cite PMID: 25998712

Line 41, Page 1 of 28: The authors should cite PMID: 28240776

Answer No.1) According to the Reviewer’s suggestion, we cited those papers in our revised manuscript..

Query No.2)The authors should prepare Abbreviations part (with order in alphabet) for this manuscript, hence, the Readers can be easier to follow also.

Answer No.2)Thank you for your suggestion. We added the abbreviations part in the revised manuscript. 

List of Changes Made

1. Page1, line 39: a new reference [3] was added.

2. Page1, line 42: a new reference [4] was added.

3. page14, line 454-481: the abbreviations part was added as follows; “6. Abbreviations

microRNA (miRNA), noncoding RNA (ncRNA), messenger RNA (mRNA), RNA polymerase II (Pol II), trans-activation-responsive RNA-binding protein (TRBP), argonaute (AGO), RNA-induced silencing complex (RISC), intraluminal vesicles (ILVs), multivesicular bodies (MVBs), microvesicles (MVs), Toll-like receptor (TLR), IL-1R-associated kinase 1 (IRAK1), TNFR-associated factor 6 (TRAF6), TAK1-associated binding protein 2 (TAB2), vascular cell adhesion molecule 1 (VCAM-1), microphthalmia-associated transcription factor (MITF), macrophage colony-stimulating factor (M-CSF), receptor activator of nuclear factor kappa-B (RANK), RANK ligand (RANKL), nuclear factor I/A (NFIA), mesenchymal stem cells (MSCs), Runt-related transcription factor 2 (Runx2), CCAAT-enhancer-binding proteins (C/EBPs), proliferator-activated receptor γ (PPARγ), extracellular vesicles (EVs), bone marrow-derived MSCs (BM-MSCs), colorectal cancer cells (CRC), breast cancer (BC), Osteosarcoma (OS), vascular endothelial growth factor (VEGF), VEGF receptor (VEGFR), microvessel density (MVD), hypoxia inducible factor 1 α (HIF1α), erythroblastic leukemia viral oncogene homolog (ERBB), mitogen-activated protein kinase (MAPK), depletion of cell division cycle 5-like (CDC5L), cyclin-dependent kinase 4 (CDK4), mouse double minute 2 homolog (MDM2), matrix metalloproteinase (MMP), tyrosine kinase (RTK), dickkopf-related protein 1 (DKK1), Li–Fraumeni syndrome (LFS), endoplasmic reticulum (ER), transforming growth factor-β (TGF-β), cancer stem cells (CSCs), diallyl trisulfide (DATS), alkaline phosphatase (ALP), event-free survival (EFS), areas under the curve (AUC), calneuron1 (CALN1), multidrug resistance 1 (MDR-1), P-glycoprotein (P-gp), cancer-associated stromal fibroblasts (CAFs), chondrosarcoma (CS), mechanistic target of rapamycin (mTOR), SRY-related high mobility group box gene (SOX), chemokine C-C motif receptor 5 (CCL5), WNT1-inducible signaling pathway protein-3 (WISP-3), G-protein signaling 16 (RGS16), CXC chemokine receptor 4 (CXCR4), basic fibroblast growth factor (bFGF), brain-derived neurotrophic factor (BDNF), Ewing’s sarcoma (EWS), insulin-like growth factor (IGF), IGF receptor 1 (IGF-1R), rhabdomyosarcomas (RMSs), embryonal RMS (ERMS), alveolar RMS (ARMS), liposarcoma (LPS), well-differentiated liposarcoma (WDLPS), dedifferentiated liposarcoma (DDLPS), myxoid liposarcoma (MLPS), pleomorphic liposarcoma (PLPS), casein kinase 1α (CK1α), homeobox protein A5 (HOXA5)”.

4. Page15, line 500,501: New reference was added as follows; [3], “Lin, S.; Gregory, R.I. MicroRNA biogenesis pathways in cancer. Nat. Rev. Cancer 2015, 15, 321–33.”.

45 Page15, line 502,503: New reference was added as follows; [4], “Bach, D.-H.; Hong, J.-Y.; Park, H.J.; Lee, S.K. The role of exosomes and miRNAs in drug-resistance of cancer cells. Int. J. cancer 2017, 141, 220–230.”.